# Could a Non-Cellular Molecular Interactome in the Blood Circulation Influence Pathogens’ Infectivity?

**DOI:** 10.3390/cells12131699

**Published:** 2023-06-23

**Authors:** Eugenio Hardy, Hassan Sarker, Carlos Fernandez-Patron

**Affiliations:** 1Center of Molecular Immunology, P.O. Box 16040, Havana 11600, Cuba; 2Department of Biochemistry, Faculty of Medicine and Dentistry, College of Health Sciences, University of Alberta, Edmonton, AB T6G 2H7, Canada; hsarker@ualberta.ca

**Keywords:** nanoparticles, virus, bacteria, protein corona, glucocorticoid, innate immunity

## Abstract

We advance the notion that much like artificial nanoparticles, relatively more complex biological entities with nanometric dimensions such as pathogens (viruses, bacteria, and other microorganisms) may also acquire a biomolecular corona upon entering the blood circulation of an organism. We view this biomolecular corona as a component of a much broader non-cellular blood interactome that can be highly specific to the organism, akin to components of the innate immune response to an invading pathogen. We review published supporting data and generalize these notions from artificial nanoparticles to viruses and bacteria. Characterization of the non-cellular blood interactome of an organism may help explain apparent differences in the susceptibility to pathogens among individuals. The non-cellular blood interactome is a candidate therapeutic target to treat infectious and non-infectious conditions.

## 1. Introduction

Any atomic/molecular assembly (nanoparticles, vaccine constructs) or pathogen (virus, bacteria, fungi, parasite) that is introduced to the blood circulation of an organism interacts with cellular and non-cellular (extracellular) components of the blood. In this contribution, we suggest that the non-cellular components of the blood define an interactome that may serve functions in innate immunity. 

The innate immune system is an ancient form of host defence that has evolved over time to protect organisms from infection and damaged self [1,2]. There are two main types of innate immunity: constitutive and inducible. Constitutive innate immunity is considered the first line of defence against foreign pathogens and danger signals and includes physical barriers, cellular components (e.g., macrophages), and other components such as natural antibodies, acute phase proteins, the complement system, and the coagulation cascade [2]. Constitutive defences share common features, such as their manifestation at sites of constant interaction with pathogens, their destructive potential focused exclusively on the stimuli (e.g., microbes)—not on host cells or tissues, and their lack of potential to augment the innate immune response [3,4]. Most endogenous defence mechanisms are activated by infection, but some need specific recognition of the pathogen [3,4,5]. Inducible innate immunity generally involves pathogen-associated molecular patterns which are highly conserved molecular structures produced only by microorganisms (such as pathogenic microbes), but not by the host [3,4,5]. Upon recognition of pathogen-associated molecular patterns by receptors in cells of the innate immune system known as pattern-recognition receptors, a variety of signals such as costimulatory molecules, inflammatory cytokines, and chemokines are released that mediate a series of antimicrobial immune responses. These signals also activate and instruct the generation of pathogen-specific and sustained adaptive immune responses against persistent pathogens [3,4,5]. Innate mechanisms dependent on inducible pattern-recognition receptors can lead to very potent and effective protective responses, but also to excessive inflammatory and immunopathological reactions [5]. Although a wealth of published data exists on inducible innate responses mediated by pattern-recognition receptors that protect individuals from viral, bacterial, and parasitic infections, much remains to be understood about the constitutive immune mechanisms [5].

Here, we develop the hypothesis that pathogens in the blood circulation encounter a non-cellular molecular interactome that may include large and small biomolecules (proteins, peptides, metabolites) which can be recruited to the pathogens’ surface (much like a biomolecular corona). We postulate that this interactome may modulate pathogens’ infectivity. We argue that the non-cellular blood interactome is likely specific to the host and capable of modifying the infectivity potential of pathogens; its characterization could help explain the differences in susceptibility to pathogens’ infection among individuals as well as improve the effective clinical management of infectious and non-infectious conditions.

The structure of the article is as follows: First, we review data suggesting the rapid recruitment of layers of biomolecules (biocorona) that avidly interact with the surface of artificial nanoparticles when these are introduced to biological fluids. This leads us to review evidence suggesting the recruitment of biocorona analogues to the surface of viruses, which are relatively complex molecular assemblies of nanometric dimensions. As an example, we review findings from our laboratory suggesting that glucocorticoids may form a biocorona through their ability to bind to multiple pockets on Spike, the surface protein of SARS-CoV-2 (severe acute respiratory syndrome-coronavirus-2)—the causative agent of the 2019 coronavirus disease pandemic (COVID-19) [6]. We then propose that, much like artificial nanoparticles and viruses, the biological impact of bacteria and other microorganisms invading the blood circulation of an organism may be influenced by the non-cellular blood interactome, which effectively results in a pathogen-acquired biocorona. A simple bait–prey thermal proteome profiling technique is proposed to screen for molecules in the blood circulation that bind to components of pathogens. Section 2 closes this contribution with the proposal of a novel bridge between the biochemistry and biology of artificial nanoparticles in the blood circulation and those of relatively complex biological entities with nanometric dimensions such as pathogens (viruses, bacteria, and other microorganisms). Several key issues that need to be addressed in future research in this area are listed and briefly commented on in Section 3.

### 1.1. Artificial Nanoparticles and Their ‘Acquired Protein Coronas’

The term nanoparticle refers to any natural or artificial structure, including tubes and fibers, with external dimensions between 1 and 100 nanometers (nm) [7,8,9]. A spectrum of technologies enables nanoparticles to be manufactured and engineered with a specific synthetic (material-intrinsic) identity including rational coating of their surface to facilitate specific applications including tissue targeting in biomedical applications [10,11,12,13] (see Table 1, column 2). For example, nanoparticles that have been covalently linked to macromolecules such as polyethylene glycol, antibodies, or peptides are designed to ensure desirable levels of solubility, stability, or biological activity in drug delivery (Table 1, columns 1–3).

One of the major aims in manipulating the surface properties of nanoparticles is to control their targeting and pharmacokinetics properties which are necessary for efficacious drug delivery [24,25]. When exposed to a physiological environment, such as the blood, nanoparticles interact with components of their environment. It is known that proteins in the blood convert naked nanoparticles into nanoparticles with a surface protein coat called a “protein corona” [26,27,28,29] (see Figure 1). The formation of a protein corona on nanoparticles is mediated by mutual electrostatic and hydrophobic interactions, hydrogen bonding, van der Waals forces, or π-π stacking. The protein corona confers a new physical, chemical, and biological identity to the nanoparticles [28,29,30,31,32,33,34,35,36].

There is no universal protein corona; rather, each type of nanoparticle in a given environment acquires (i.e., adsorbs) a protein corona whose composition depends on the particle—and the environment. Table 1 (column 4) illustrates the diversity of proteins that can adsorb to nanoparticles.

A protein that has been adsorbed to a nanoparticle can desorb from it or be replaced by another protein with a higher affinity [27,28,37]. The composition of the protein corona may thus evolve over time. A recent model postulates that the protein corona can form in a few seconds [27,31]. Further, nanoparticles carry proteins from one biological environment to another on their protein corona [27,31].

Much of the biological behavior of nanoparticles, including aggregation, circulation time, clearance rate, and targeting, are greatly influenced by the protein corona [10,26,30,32,38,39], which may include proteins involved in blood coagulation, immune responses including complement activation, and lipid transport (Table 1, column 4). Protein coronas can concentrate proteins known as opsonins, such as complement proteins, coagulation proteins, and antibodies. Opsonins stimulate immune cell recognition and facilitate rapid clearance of nanoparticles from the bloodstream by immune cells such as macrophages [17,39,40,41,42,43]. Opsonic properties are also shared by proteins involved in lipid transport such as apolipoprotein A-I and apolipoprotein E [41]. Alpha-2-HS-glycoprotein can act as an opsonin, enhancing macrophage-deactivating mechanisms [44]. Apolipoproteins or other proteins with dysopsonin-like properties, such as histidine-rich glycoprotein, clusterin (apolipoprotein J), and albumin, which are present in protein coronas of nanoparticles (Table 1, column 4), may prolong the time that nanoparticles circulate in blood [41].

Interestingly, protein coronas have also been identified on nanoparticles whose surface has been chemically modified, e.g., through the attachment of hydrophilic polymers such as polyethylene glycol (Table 1, rows 3, 4, and 7), which is useful to lengthen blood circulation times and to decrease the hepatic uptake of the nanoparticles. Protein coronas have been described for nanoparticles both in vivo (as discussed) as well as in vitro (Table 1).

Nanoparticles in biological environments including saliva, lung fluid, and blood have been found to interact with phospholipids, carbohydrates, nucleic acids, and other metabolites, in addition to proteins and peptides [10,45,46]. As a result, a layer of biomolecules (biocorona) may end up coating the nanoparticles’ surface giving nanoparticles a new biological identity [26,30,31,32,33,36,47]. Ultimately, the biocorona mediates interactions between the nanoparticle and the host biological system [15,30,31,32,33,34,48].

### 1.2. Nanoparticles Can Acquire Biocoronas: Do Viruses also Acquire a Biocorona?

Viral particles can have diverse shapes, including helical and icosahedral, and nanometric sizes (typically ranging from 20 to 300 nm) [49]. Apart from a nucleocapsid made of identical proteins (capsomers) enclosing the viral genome that can be RNA or DNA, viruses can have an envelope structure with phospholipids taken from host cells and pH-dependent surface charge [50]. In non-enveloped viruses, a coat protein controls the viral surface charge [51]. 

The recruitment of biocoronas described for artificial nanoparticles may be pervasive and affect the physico-chemical and biochemical potential of many structures, not just nanoparticles. Much like artificial nanoparticles in extracellular environments, viruses can attract layers of ions and molecules to their surface, including large biomolecules (e.g., proteins, peptides) and small biomolecules (e.g., metabolites such as lipids, steroids, or saccharides) that form a viral biocorona capable of influencing viral infectivity (Figure 2). In fact, phosphate and calcium ions from the surrounding intra- and extra-cellular media can bind to moieties on the viral surface, potentially changing the isoelectric point and overall surface charge of viral particles [51]. Antimicrobial peptides such as defensins (e.g., natural human neutrophil defensin 1-3, human defensin 5, and human β-defensin 3) can interact with glycoproteins from herpes simplex virus-1 and -2, thus affecting viral entry into host cells [52,53,54]. Distinct acquired coronas are displayed by different coronavirus variants [55]. 

The acquisition of a biocorona is expected to influence the viral particles’ behavior outside of host cells as well as virus–host cell interactions [55]. Rich and unique acquired biocoronas have been documented in herpes simplex virus type 1 and respiratory syncytial virus in various biological fluids to yield viral particle populations ranging from tens to a few hundreds of nanometers in size and diverse biocorona profiles [63]. Conceivably, these viral particles can be endocytosed by macrophages. Complement factors, properdin, protein S100, vimentin, and annexin A1 have been found in protein coronas of respiratory syncytial virus and may contribute to viral neutralization and/or influence viral infectivity as well as play roles in the modulation of the host immune response to the virus [63]. 

### 1.3. Cortisol and Dexamethasone Can Bind to Multiple Sites on SARS-CoV-2 S1: Could Glucocorticoids Be Components of Viral Biocoronas?

SARS-CoV-2 viral particles are approximately 60–140 nm in diameter and infect human cells through binding of the viral Spike protein (which is approximately 9 to 12 nm in length) to receptor proteins on the host cell surface [65]. Spike comprises two subunits, S1 and S2, which share an extracellular domain. The S1 subunit includes an N-terminal domain and receptor binding domain (RBD) which allow these coronaviruses to bind to host angiotensin converting enzyme 2 (ACE2 receptor) during virus entry into cells [66]. Clusters of differentiation 147 and 26 (CD147 and CD26 receptors) and transmembrane protease serine 2 (TMPRSS2 co-receptor) also participate in enabling the viral infection process [66,67].

SARS-CoV-2 can trigger the hypothalamic–pituitary–adrenal axis, leading to increased secretion of glucocorticoids from the adrenal cortex [6]. Once released, glucocorticoids circulate in the blood and contribute to the regulation of inflammatory signaling, systemic immune responses, and the metabolism of lipids and carbohydrates [68]. By binding to cytosolic/nuclear receptors, glucocorticoids can exert their effects through signal transduction pathways [68,69,70]. 

In contrast to the traditional view of the effects of glucocorticoids on cellular immunity, which is complex and cell-type specific [71,72], a new non-traditional view of the effects of endogenous glucocorticoids would be that constitutively secreted and SARS-CoV-2-induced glucocorticoids may be available to interact with viral components. Starting with *in-silico* studies (docking and molecular dynamics), we identified and validated unique pockets in SARS-CoV-2 S1 available for high-affinity binding of cortisol to S1, and concentration-dependent inhibition of the S1-ACE2 interaction [73]. These binding pockets are situated and distributed across the RBD, N-terminal domain, RBD–RBD interface, and N-terminal domain–RBD interface. We used limited proteolysis coupled to liquid chromatography–mass spectrometry to confirm several of the cortisol-binding pockets identified by molecular dynamics (e.g., HCY_8, HCY_29, HCY_35, HCY_59, HCY_88, HCY_112, HCY_153, and HCY_161) and we determined the amino acid sequences to which cortisol binds. We corroborated these data with a cortisol-acetylcholinesterase conjugate assay, which showed that S1 can bind and scavenge free cortisol in solution and that binding of cortisol to S1 causes denaturation of S1, as shown using a GloMelt™ Thermal Shift Protein Stability [73]. Moreover, we found that nanomolar concentrations of cortisol inhibited the interaction between S1 and ACE2 [73]. Cortisol (100 nM) inhibited the interaction between the SARS-CoV-2 S1 Beta variant (E484K, K417N, N501Y) and ACE2 by ~55% inhibition. In contrast, some mutations in the Delta and Omicron variants of concern are located in or in the vicinity of cortisol-binding pockets and may reduce the effectiveness of cortisol binding to these variants of S1 [73]. As Spike mutations affecting cortisol binding to SARS-CoV-2 S1 could increase SARS-CoV-2 infectivity, cortisol interactions with S1 could be important for viral infectivity [6,73]. 

We have proposed that the interactions between endogenous glucocorticoids and viral components such as S1 in SARS-CoV-2 may present a potentially novel innate immunity mechanism, through which glucocorticoids could participate in directly reducing viral infectivity [6,73]. A question that has not yet been experimentally addressed is whether there are other constitutive or viral-induced molecules in the human blood such as peptides and non-antibody proteins which define a non-cellular interactome cognate to structural components of SARS-CoV-2 (including S1) and capable of influencing SARS-CoV-2 infectivity either directly or in concert with glucocorticoids, such as cortisol.

When it comes to interactions with host factors, many viruses (including coronaviruses other than SARS-CoV-2) display similarities with artificial nanoparticles in that both can attract and become decorated by host factors found in extracellular environments (Figure 3). This should be unsurprising given the nanometric dimensions of viral particles. 

In particular, the coronavirus particles, which range in size from 60 to 120 nm, have a viral surface made of a lipid bilayer (~85 nm in diameter) that is embedded with structural glycoproteins, including the membrane, envelope, and spike structural proteins [76,77]. Spike proteins, the component responsible for the surface’s crown-like appearance of coronaviruses (i.e., so named for their ‘inherited’ corona), are typically 20 nm long [78]. As with artificial nanoparticles, host proteins may be attracted to the surface of coronaviruses forming biomolecular coronas (‘acquired’ biocoronas) [55] (Figure 3). These acquired biocoronas can be as diverse as the dissimilar tissue microenvironments in which they develop and their different molecular compositions [55]. Mixed acquired biocoronas might result from the replacement of some molecular components of an initial biocorona with other biomolecules found in areas where coronaviruses are found, such as the circulation or body tissues [55]. Conceivably, the acquired corona can influence the interaction between the coronavirus and the host interrupting coronavirus binding through unconventional lung cell receptors, disrupting the lysosome’s capacity to break down invasive coronaviruses, biodistributing coronaviruses in various tissues, stimulating immune responses, and altering symptoms brought on by the coronavirus, as described for SARS-CoV-2 variants with altered acquired biocorona [55]. One example is human serum albumin, whose interactions with SARS-CoV-2 S1 may block antigenic sites on the RBD of S1, thus interfering with neutralizing antibodies with affinity for the RBD [79,80]. Another example is glucocorticoids. SARS-CoV-2 has 52 high-affinity glucocorticoid binding pockets on S1; the binding of cortisol to multiple sites on S1 decreases S1 affinity for ACE2 and may thus influence infectivity and disease severity, as we have proposed [6,73]. Conceivably, other molecules in the blood with affinity for S1 could influence (positively or negatively) cortisol affinity for S1 as well as S1 affinity for ACE2. This influence may be as diverse as competitive binding to cortisol pockets on S1, non-competitive binding to S1 regardless of whether cortisol is already bound or not, or non-competitive binding to preformed cortisol/S1 complexes. As we have proposed for SARS-CoV-2 [6,73], glucocorticoids may exert influences on the infectivity potential of other coronaviruses that use Spike protein for infection. A similar reasoning may be applied to describe the probable impact of an acquired biocorona on members of other families of viruses. 

Understanding how the non-cellular (non-antibody) blood interactome of a virus affects the virus–host interactions merits investigation. We anticipate the non-cellular blood interactome to be responsible, at least in part, for differences in the susceptibility to viral infections among individuals as well as being a new candidate therapeutic target to improve the treatment of viral conditions.

### 1.4. Nanoparticles and Many Different Viruses Can Acquire Biocoronas: Do Pathogenic Bacteria and Other Microorganisms Acquire a Biocorona?

Analogous to the notions described previously for nanoparticles and viruses (Figure 1, Figure 2 and Figure 3), we postulate that any bacterium that invades the blood circulation has the potential to encounter a non-cellular blood interactome, which can influence the bacterium–host interactions (Figure 4).

Bacterial cells, which range in size from 0.15 to 700 μm, can be thought of as colloidal particles [86]. The prokaryotic cell membrane is rich in cardiolipin and phosphatidylglycerol, which are glycerophospholipids with a net negative charge [87]. Teichoic and lipoteichoic acids as well as lipopolysaccharides are additional anionic membrane components on Gram-positive bacteria and Gram-negative bacteria, respectively [87]. The resultant negatively charged surface of many bacteria is known to attract positively charged antimicrobial peptides in blood, tears, saliva, and urine [88,89]. Depending on concentration, secondary structure, and physical-chemical characteristics (surface charge, hydrophobicity, and stability) of the lipid membrane, antimicrobial peptides can increase bacterial membrane permeability or result in structural damage followed by bacterial death [88,89,90,91,92]. In addition to antimicrobial peptides, a variety of antimicrobial proteins produced by tissues and innate immune cells are attracted to negatively charged surfaces of bacterial pathogens. Some of these host proteins serve as pattern-recognition receptors (extracellular soluble pattern-recognition molecules) for components on pathogenic bacteria [93]. Examples of these pattern-recognition molecules are conserved multimeric proteins called pentraxins (e.g., C-reactive protein and serum amyloid P), lectins such as collagen-like lectins known as collectins (e.g., mannose-binding lectin and surfactant protein A) and ficolins, and other complement molecules such as the complement component 1q and complement component C3b. Ficolin molecules contain a fibrinogen-like domain in addition to a collagen-like domain, and this domain has a particular affinity for N-acetylglucosamine [94].

Soluble pattern-recognition molecules in extracellular fluids are known to interact with pathogen-associated molecular patterns such as bacterial surface glycolipids and glycoproteins [91]. For instance, host C-reactive protein binds with high affinity to phosphocholine linked to bacterial polysaccharides as well as to phospholipids and glycans [95]. Serum amyloid P binds to phosphorylcholine, phosphatidylethanolamine, lipopolysaccharides, and bacterial surface sugars (e.g., galactose, mannose) [96,97,98]. Mannose-binding lectin binds to phospholipids, carbohydrates (mannose, N-acetylglucosamine), and non-glycosylated proteins [99]. Surfactant protein A binds N-acetyl mannosamine and bacterial phospholipids [100]. Collectins/ficolins bind to carbohydrate moieties displayed on bacterial surfaces through their carbohydrate-recognition (lectin) domains [94]. Ficolins typically bind to acetylated polysaccharides, including N-acetylgalactosamine and N-acetylglucosamine, and they may also interact with bacterial peptidoglycan, lipopolysaccharides, and sialic acid [101]. Complement component 1q can bind directly to pathogenic bacteria as well as to antigen–antibody (IgM, IgG1, and IgG3) complexes found on the bacteria surface [102,103]. C3b is a stable fragment derived from complement component 3 that covalently binds to bacterial lipopolysaccharides [104]. These well-known examples illustrate how the non-cellular blood interactome affects bacterial pathogens leading to the equivalent of an ‘acquired’ biocorona on the bacterial surface. There is a wealth of knowledge on the impact of bacteria–host interactions on bacterial pathogenicity. Soluble pattern-recognition molecules contribute to host innate immune defence mechanisms, such as complement activation and opsonization, which facilitate uptake by phagocytes, bacterial neutralization, and inflammation control. The discussion of these mechanisms is out of the scope of this essay paper, but there are many excellent and comprehensive reviews on the topic [105].

Translating our earlier discussions on nanoparticles and viruses to bacteria, it is possible to make the point that bacteria invading the blood circulation of an organism encounter a non-cellular blood interactome whose interaction with bacterial surface components could influence bacterial attachment (to host cells), colonization (of tissues), and (intracellular) invasion [106,107] (Figure 4). Through disrupting or weakening of the bacterial invasive capabilities, the blood interactome could facilitate the removal of bacteria through the body’s (conventional) innate responses.

## 2. Conclusions

We postulate that the non-cellular blood interactome is a branch of the innate immune response through which organisms with a circulatory system respond to external entities as varied as any atomic/molecular assembly (nanoparticles, vaccine constructs) or pathogen (virus, bacteria, fungi, parasite). The non-cellular blood interactome is an intimate part of the pathogen interaction with the host that is likely to be highly individual-specific (e.g., due to genetic background, sex, gender, age, ethnicity, socio-epidemiological environment, previous exposure to pathogens, and presence of comorbidities). Characterization of the individual-specific non-cellular blood interactome could help explain the differences in predisposition to the pathologies elicited by pathogens in different individuals.

Whereas we postulate the non-cellular blood interactome establishes direct interactions with the invading pathogen, we do not neglect that pathogens will invariably activate multiple innate responses such as antimicrobial peptides (associated with destabilization/disruption of membrane integrity and inhibition of pathogen growth), soluble lectins (mediators of host defences), and basal autophagy for bacterial degradation [5].

Our notions are experimentally testable. There are proteomics and biochemical approaches that can be applied to identify components of the non-cellular blood interactome of pathogens. In Appendix A, we describe a novel bait–prey thermal proteome profiling technique combining liquid chromatography–mass spectrometry to identify SARS-CoV-2 S1 binders in the blood. This bait–prey interaction proteomics approach can, in combination with pathogen-specific infectivity assays, facilitate the rapid identification of pathogen-specific molecular interactors in the blood. 

The notions presented here could catalyze the design of new approaches targeting the individual-specific blood interactome to improve the treatment of old and emerging pathogens as well as non-infectious conditions.

## 3. Outlook

### Several Questions Merit Investigation in Future Studies

Is it non-cellular molecular recruitment or immune cell recognition that occurs first when a pathogen enters the bloodstream? In analogy to artificial nanoparticles, we have postulated that pathogens acquire a biocorona as they invade, circulate, and reside in the host. It is plausible that this biocorona depends on the point of entry of the pathogen, i.e., depending on whether the pathogen invades the host through the blood, airways, or gastrointestinal system, which will affect the biomolecular composition of the acquired corona [108,109,110]. Focusing on the non-cellular blood interactome as a major source of the biocorona components, the order and rate at which these components are adsorbed onto the pathogen surface warrants investigation. With silica nanoparticles in human plasma [17], protein adsorption is detected in a few seconds; it is conceivable that a similarly fast rate of adsorption will be observed with pathogens. The development of the innate immune response is also rapid, typically within minutes to hours [111]. We think that the non-cellular blood interactome should be viewed as an innate immune response that can impact the biological actions of pathogens and non-pathogenic invading entities (e.g., vaccine constructs) alike. What is the spatial architecture of the pathogen-attracted biomolecular corona? Future research may reveal whether distinct layers of biomolecules interact with the inherited pathogen biocorona, as it has been postulated that artificial nanoparticles may be covered by a (soft) protein corona located on top of an inner (hard) protein corona directly bound to the nanoparticle surface [32]. We postulated that the pathogen–protein interaction is established rapidly, possibly in a few seconds, as proposed by Tenzer et al. for artificial nanoparticle–protein interactions, and that the resulting pathogen interactome does not change in composition, but only in the quantity of proteins taken up over time [27]. Those are areas warranting research as the proteins on the corona are susceptible to post-translational modifications, including the binding of lipids and carbohydrates.

The non-cellular blood interactome is likely to be host-specific and capable of modifying pathogen infectivity. How does this explain individual susceptibility to disease? Currently, it is impossible to answer this question. Blood protein composition may be influenced by innate factors such as age, nutritional status, genetics, immune competence, and underlying chronic diseases, and by extrinsic factors such as drug use, lifestyle, and geographic origin [112]. These factors could influence the composition of the host-specific non-cellular interactome of the pathogen. Changes in protein corona composition have been documented with artificial nanoparticles [112,113]. Proteins’ conformation may change in many diseases [113], potentially altering the blood interactome of the pathogen. Like the proposal of a “personalized” protein corona which affects nanoparticles’ biochemistry and biology [112,113], we anticipate that the pathogen enters the host and circulates in association with a host-specific (“personalized”) non-cellular blood interactome. Hypothetically, the blood proteome and metabolome may differ in different disease states consistent with a state-specific blood interactome that forms the basis for the state- and host-specific acquired pathogen’s biocorona. State-specific post-translational modifications of blood proteins may affect the pathogen-associated non-cellular interactome. Proteins surrounding the pathogen surface may undergo misfolding and aggregation (protein unfolding has been reported in studies with artificial nanoparticles [114]). Pathogen–protein interactions may induce the exposure of hidden protein epitopes (as happens with nanoparticle-denatured proteins [28,114]). This could enhance immunological responses and/or induce unwanted inflammatory responses. We postulate that the host-specific non-cellular blood interactome is an important external variable that can render the host susceptible or resilient to infectious diseases. Recognition of the non-cellular blood interactome as an innate immune response could help explain why an infectious agent that is virulent in one setting may be mild or harmless in another. 

Do pathogens containing biologically active molecules differ from artificial nanoparticles in the way they form protein coronas? Bissantz et al. and Zhou et al. have discussed in great detail a variety of non-covalent chemical interactions that may exist between two partner molecules in biological systems, including van der Waals contact (less than 1 kcal/mol), hydrophobic force (1.5–2 kcal/mol), salt bridge (0.5–5 kcal/mol), π-π stacking (5–7 kcal/mol), electrostatic interaction (below 20 kcal/mol), hydrogen bonding (0.25–40 kcal/mol), and halogen bonding (1–40 kcal/mol) among others [115,116]. Hydrogen bonding, halogen bonding, salt bridge, and π-π stacking are enthalpy-driven, strong interaction forces [116]. Halogen bonds are considerably weaker than hydrogen bonds. The gains in binding affinity they produce, however, can be substantial [115]. Hydrogen bonding and halogen bonding are highly specific forces [116]. Salt bridge and π-π stacking are moderate specific forces [116]. Hydrophobic and van der Waals forces are weak in strength and have a low degree of specificity [116]. The hydrophobic force is driven by entropy and the van der Waals force is driven by enthalpy [116]. These interactions typically drive the formation of protein coronas on artificial nanoparticles [116]. 

We hypothesize that similar biomolecular interaction types enable the formation of an acquired biocorona on pathogens and artificial nanoparticles. This latter notion could be testable for the case of pathogens through approaches analogous to those previously applied to nanoparticles [30]. In vitro experiments have been useful in demonstrating that the ability of artificial nanoparticles to target cells, internalize cells, and cause cytotoxicity differs significantly depending on whether a protein corona is present or absent [117]. In vitro studies cannot mimic the effects of inflammatory responses activated in the body after nanoparticle delivery [30,117]. Previous research on magnetic nanoparticles showed that these particles attract a different proteome under in vivo conditions than under in vitro conditions [30,118]. To better mimic in vivo conditions and determine the acquired (biological) identities of pathogens, ex vivo studies could be performed with freshly prepared whole blood [119].

Do post-translational modifications of the pathogen-attracted interactome affect innate immune responses to the pathogen? Answering this question would require detailed studies at the proteomics and metabolomics levels. Among the candidate post-translational modifications of biocorona components, lipid as well as sugars are commonly attached to proteins. In particular, sugars are attached to proteins by glycosylation (the enzymatic attachment of carbohydrates) or glycation (the non-enzymatic attachment of monosaccharides) [120] as many blood proteins are glycosylated or glycated [121,122]. Glycosylation of immunoglobulin G (i.e., the enzyme-mediated addition of sugar molecules to IgG) is involved in humoral immune responses in aging, inflammation, responses to cancer cells, pathogen infections, and autoimmune diseases [120]. Diabetes can lead to glycation of blood proteins [123]. It is plausible that the non-cellular interactome associated with pathogens involves proteins modified by either glycosylation or glycation. These modifications may affect (as well as being biomarkers of) the host susceptibility to severe disease induced by the pathogen. 

Does the composition of the non-cellular blood interactome that associates with a pathogen change as a function of the pathogen entry route? The composition of proteins and other biological molecules varies among biological fluids. The biocorona acquired by pathogens is likely to vary among the point of invasion and the biological fluid(s) to which the pathogen encounters during invasion, circulation, and residence in the host. For example, the human nasal secretions proteome differs in structure and post-translational modifications from the human blood proteome [108]. While nasal secretions are made up of plasma components and serous fluid, they also contain mucus and secretions from distinct epithelial and immune cells [108]. Gastrointestinal fluids contain diverse carbohydrates, phospholipids, and mucin expressed at levels not typical of the human blood [110]. The fluid that lines the human airways (pulmonary surfactant) is also different from non-cellular interactome featured in human blood [109]. Biomolecules present in biological fluids are potential interactors of artificial nanoparticles and pathogens alike and represent an extension of the biomolecular repertoire that is likely to constitute the biocorona predicted to associate with pathogens. Noteworthily, most earlier studies on nanoparticle biocorona have been conducted with blood proteins and very infrequently with other biological fluids [30]. Our general hypothesis is that pathogens attract non-cellular interactomes from biological fluids (blood in particular) which depend on where or how they invade the host. A similar hypothesis has been proposed for the specific case of coronaviruses whose biocorona might differ depending on entry routes and environments (e.g., blood, nasal, oral, and respiratory mucosal tissues) [55].

## Figures and Tables

**Figure 1 cells-12-01699-f001:**
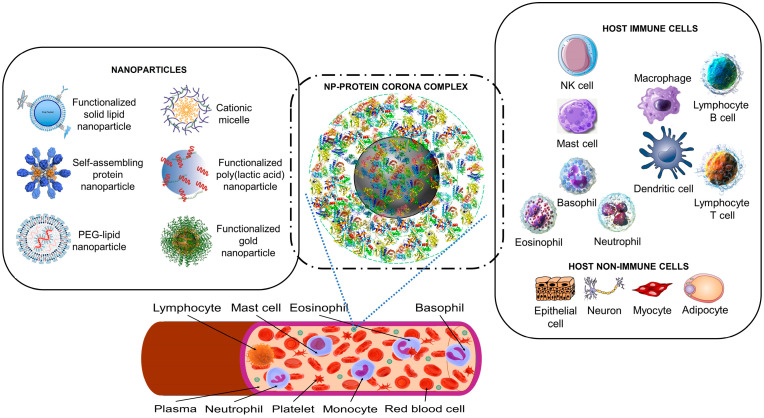
The nanoparticle protein corona interacts with host cells. Synthetic nanoparticles (on the left) enter the bloodstream, where they interact with proteins (a proteomic non-cellular blood interactome) to form a nanoparticle (NP)–protein corona complex (middle panel). The protein corona gives the synthetic nanoparticle a biological identity that mediates interactions with innate immune cells and non-immune cells (right panel). The protein corona can also effectively hide surface ligands, reducing the interaction with intended target cell receptors [30]. The protein corona–host cell interaction activates biological responses, including innate immune recognition and clearance by phagocytic cells, adaptive immune response, therapeutic activity, and toxic reactions such as thrombocyte activation, hemolysis, and excessive pattern-recognition receptor-mediated inflammation [17,30,31,32,33,34]. Immune cells may recognize exposed corona proteins as antigens and trigger undesirable immune reactions [30,35]. Image files used to partly generate the figure are licensed under https://creativecommons.org/licenses/by/4.0 (I53-50 nanoparticle vaccine schematic.png by J. Marcandalli et al. (accessed on 1 April 2023), Vaccines-09-00065-g001.webp by M.D. Buschmann et al. (2021), Delivery methods for HIV mRNA vaccine.png by Z. Mu et al. (accessed on 5 April 2023), and Mechanisms_by_which_bacteria_target_tumors.svg by M. T-Q. Duong et al. (accessed on 7 April 2023)), https://creativecommons.org/licenses/by-sa/4.0 (DNA-AuNP0022.jpg by Shpetrosko (accessed on 4 April 2023), Giemsa_Stain_Macrophage_Illustration.png by Noah Smith (accessed on 18 April 2023), Final_stem_cell_differentiation_(1).svg by Haileyfournier (accessed on 14 April 2023), 3FFN_background_removed.png by Pronchik (accessed on 6 April 2023), and Antibody_IgG1_structure.png by Tokenzero (accessed on 8 April 2023)), https://creativecommons.org/licenses/by/3.0 (Blausen 0909 WhiteBloodCells.png by Blausen.com staff (accessed on 14 April 2023)), https://creativecommons.org/licenses/by-sa/3.0 (SolidLipidNanoparticle.jpg by Andrea Trementozzi (accessed on 8 April 2023), Cardiovascular system-Lymphopoiesis-NK cell—Smart-Servier.png by Laboratoires Servier (accessed on 13 April 2023), and Protein_CFB_PDB_1dle.png, Protein_TF_PDB_1a8e.png, Protein_C3_PDB_1c3d.pngtructure of the C3 protein and Protein_CP_PDB_1kcw.png, all by Emw (accessed on 11 April 2023)), https://creativecommons.org/licenses/by/2.0 (Mast Cell (30107399584).jpg by NIAID (accessed on 18 April 2023)), and https://creativecommons.org/publicdomain/zero/1.0/ (PDB_1ao6_EBI.jpg by Jawahar Swaminathan, C5.png by Pietro Roversi, Antithrombin_monomer.jpeg by K. Murphy, and Lymphocyte, Mast cell, Eosinophil, Basophil, Neutrophil, Platelet, Monocyte and Red blood cell, all by Sarbasst Braian), via Wikimedia Commons. Abbreviations: PEG, polyethyleneglycol; NK cell, natural killer cell.

**Figure 2 cells-12-01699-f002:**
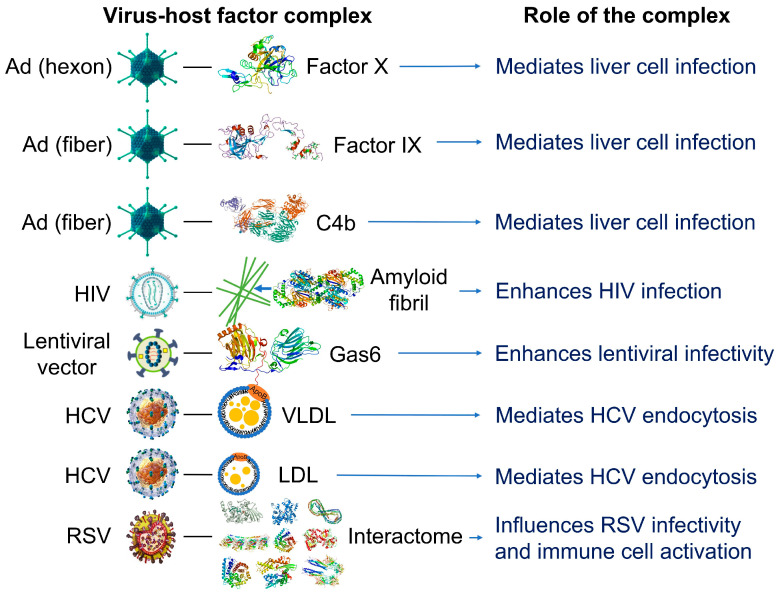
Examples of reported interactions involving viruses and components of the blood interactome (left panel). Impact of the resulting complexes on viral infectivity and host immune response (right panel). Adenovirus (Ad), a 90–100 https://en.wikipedia.org/wiki/Nanometernm virus composed of double-stranded DNA, an icosahedral nucleocapsid, and three major capsid proteins (hexon, penton, and fiber), attracts and interacts with blood factors including coagulation factors (factor IX and factor X) and complements C4B [56,57,58,59]. Human immunodeficiency virus (HIV), a single-stranded, positive-sense ribonucleic acid-containing, spherical (100–120 nm), enveloped virus that also contains a capsid core and proteins (e.g., gp120 and gp41) embedded in the lipid bilayer, attracts and interacts with fibril-forming prostatic acid phosphatase fragments [60]. Lentiviral vectors, a type of retroviruses originated from human immunodeficiency viruses, attract and interact with a gamma-carboxyglutamic acid domain-containing protein called Gas6 (growth arrest-specific 6). Lentiviral vectors pseudotyped with envelope proteins from various sources, including the Ross River virus, Sindbis virus, and baculovirus, induce this interaction [61]. Hepatitis C virus (HCV), a small, enveloped virus of 55–65 nm in diameter, consisting of positive-sense single-stranded RNA and glycoproteins (e.g., E1 and E2) embedded in the lipid bilayer, attracts and interacts with lipoproteins such as low-density lipoproteins (LDL) and very low-density lipoproteins (VLDL) [62]. Respiratory syncytial virus (RSV), an enveloped, spherical, and sometimes filamentous (150 nm) virus with a single-stranded negative-sense RNA, a capsid core, and glycoproteins (e.g., such as F, G, and SH proteins) embedded in the lipid bilayer, attracts and interacts with hundreds of host proteins depending on the extracellular biologically fluid (e.g., human plasma and bronchoalveolar lavage fluid) [63]. In all cases, the resulting acquired corona serves as the surface for actual contact with host cells and may influence the ability of viruses to activate and enter the host cells. Instead of just one protein component, the combination of corona components that are enriched on the viral surface affects viral infectivity [63]. The image files that were used to generate part of the illustration are licensed under https://creativecommons.org/licenses/by-sa/4.0 (Adenovirus_3D_schematic.png (accessed on 5 March 2023) and HI-virion-structure en.svg (accessed on 1 March 2023) by T. Splettstoesser and HCV.png by P. Znamenskiy (accessed on 4 March 2023)), https://creativecommons.org/licenses/by/3.0 (595768.fig.001.jpg by S.S. Bawage et al. (accessed on 11 March 2023)), https://creativecommons.org/licenses/by-sa/3.0 (GAPDH_with_labels.png by Vossman (accessed on 5 March 2023), Actin_with_ADP_highlighted.png by T. Splettstoesser (accessed on 3 March 2023), and Protein_F10_PDB_1c5m.png (accessed on 10 March 2023), Protein ACPP PDB 1cvi.png (accessed on 8 March 2023), Protein_GAS6_PDB_1h30.png (accessed on 12 March 2023), Protein_TUBA1A_PDB_1ffx.png (accessed on 17 March 2023), Protein C3 PDB 1c3d.png (accessed on 15 March 2023), Protein_C4A_PDB_1hzf.png (accessed on 11 March 2023), Protein_HSPA1A_PDB_1hjo.png (accessed on 16 March 2023), and Protein_FGB_PDB_1fza.png (accessed on 14 March 2023) all by Emw), and https://creativecommons.org/publicdomain/zero/1.0/ (Lentiviral_vector.png by P. Znamenskiy, PDB_1pfx_EBI.jpg by J. Swaminathan and MSD staff, PBB_Protein_APOA1_image.jpg by ProteinBoxBot at English Wikipedia, and Lactoferrin.png by Lijealso), via Wikimedia Commons. PDB entry 5JTW was used for C4b [64].

**Figure 3 cells-12-01699-f003:**
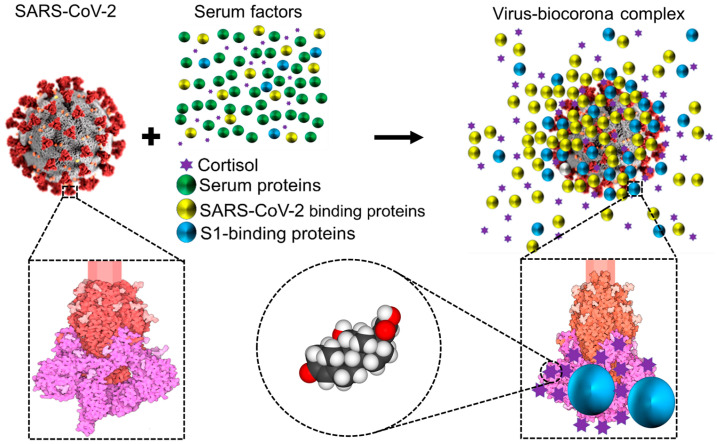
Schematic representation showing that while circulating in the extracellular environment viral particles may undergo biophysical transformations, such as becoming decorated by blood components leading to a biocorona—analogous to what has been described for nanoparticles. This notion is illustrated and has been documented for the case of SARS-CoV-2 [6,73]. Notice that the ‘yellow and blue’ (virus-binding) proteins are enriched on the virus surface, whereas the ‘green’ (non-binding) proteins are not. S1 is colored in magenta, S2 in red, and glycosylation is shown in lighter shades. Representation of the SARS-CoV Spike protein (https://cdn.rcsb.org/pdb101/motm/246/246-SARSCoV2_Spike-6crz_6vxx3.tif), which is shown magnified in this illustration, was obtained from David Goodsell (https://doi.org/10.2210/rcsb_pdb/mom_2020_6), licensed under CC BY 4.0 (https://creativecommons.org/licenses/by/4.0/). The original illustration was created using PDB entries 6crz [74] and 6vxx [75]. Abbreviations: SARS-CoV-2, severe acute respiratory syndrome-coronavirus-2; S, spike.

**Figure 4 cells-12-01699-f004:**
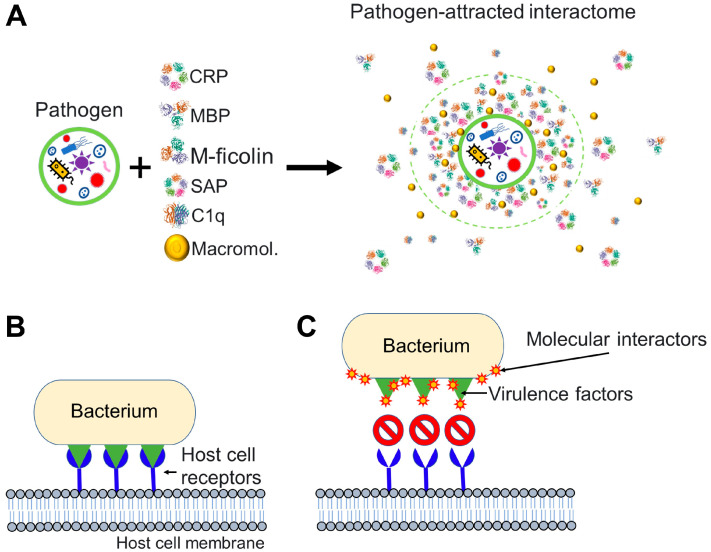
Pathogen interactions with molecules of the host can affect pathogen infectivity. (**A**) Pathogen interacts with molecular factors from the bloodstream and attracts an interactome to its surface. Components of the pathogen-attracted interactome may include pentraxins such as C-reactive protein (CRP) and serum amyloid P, collectins such as mannose-binding lectin (MBL), ficolins such as M-ficolin, complement molecules such as the complement component 1q (C1q), and many other macromolecules (Macromol.) including endogenous cortisol [6,73] and proteins and peptides. (**B**,**C**) Particular case where the invading pathogen is a bacterium. (**B**) An invading bacterium interacts with host cell receptors (in blue) through virulence factors (in green). Virulence factors are molecules that facilitate host invasion by bacterial cells. (**C**) Molecular interactome components are drawn to pathogenic bacteria and successfully prevent the interaction between the virulence factors of the bacterium and host cell receptors. To generate the figure, we used the PDB entry 1GNH for CRP [81], PDB entry 1RTM for MBP [82], PDB entry 2D39 for M-ficolin [83], PDB entry 1SAC for serum amyloid P [84], and PDB entry 1PK6 for C1q [85].

**Table 1 cells-12-01699-t001:** Diversity of proteins adsorbed to different nanoparticles ^a,b^.

NPs ^c^	Synthesis Material	Use	Top 10 Most-Abundant Constituents	Ref.
Liposomes	HSPC, DSPG, Chol	Drug delivery (e.g., liposomal amphotericin B)	Blood coagulation: Coagulation factor XIII B chain, coagulation factor XIII A chain, fibrinogen beta chain, fibrinogen gamma chain, alpha-1-antitrypsin 1-3 Adaptive immunity: Ig kappa chain C region, fibrinogen beta chain Complement pathway: C4b-binding protein Others: Serum albumin, actin (cytoplasmic 2), fibronectin	[14]
Lysolipid-containing TSL	DPPC, MSPC, HSPC, DSPE-PEG2000, Chol, Dox	Drug delivery (e.g., cancer therapy)	Acute phase: Alpha-2-macroglobulin Lipid transport: Apolipoprotein C-III, apolipoprotein E Others: Beta-globin (A8DUK0 [+2]), Beta-globin (A8DUK4), Beta-2-globin (fragment), Beta-globin OS, Alpha-globin 1 (Q91V88 [+2]), Alpha-globin A8DUV1, Ig mu chain region	[15]
PEGylated cationic liposomes	DOTAP, DC-Chol, DOPC, DOPE, and DOPE-PEG 2000	Potential vehicle to target cancer cells	Adaptive immunity: Ig kappa chain C region, Ig mu chain C region, Ig lambda-2 chain C regions Complement pathway: Complement C3, complement C1q subcomponent subunit A, complement C1q subcomponent subunit B, complement C4-B Lipid transport: Apolipoprotein C-III, apolipoprotein E Others: Serum albumin	[16]
Silica NPs modified with surface NH_2_	Silicon dioxide	Targeting drug delivery	Blood coagulation: Coagulation factor V Complement pathway: Complement C3 Complement alternate pathway: Complement factor H, complement C1r subcomponent Lipid transport: Apolipoprotein B100, apolipoprotein A Others: Fibronectin, gelsolin, thrombospondin, inter α trypsin inhibitor heavy chain H4	[17]
Negatively charged hydrophilic silica NPs	Silicon dioxide	Studies on nano–bio interfaces	Blood coagulation: Plasminogen Lipid transport: Apolipoprotein A-I Others: Serum albumin, hemoglobin fetal subunit beta, hemoglubin subunit alpha, alpha-1 antiproteinase, tetranectin, alpha-2-HS-glycoprotein, beta-2-glycoprotein 1, serotransferrin	[18]
Silica NPs bioconjugated with PEG and transferrin	Silicon dioxide, PEG, and transferrin	Active targeting	Adaptive immunity: Immunoglobulin kappa constant, immunoglobulin heavy constant mu Complement pathway: Immunoglobulin lambda-like polypeptide 5, complement C3 Lectin complement pathway: Ficolin-3 Lipid transport: Apolipoprotein A-I Others: Albumin, actin cytoplasmic 1, hemoglobin subunit beta, serotransferrin	[19]
Carbon nanotubes (ssDNA-SWCNTs)		Bioimaging, molecular sensing, delivery	Blood coagulation: Histidine-rich glycoprotein, kininogen-1, prothrombin Adaptive immunity: Ig heavy constant gamma Immunity: Haptoglobin Complement pathway: Clusterin, complement C3 Complement alternate pathway: Complement factor H, complement C1r subcomponent Lipid transport: Aapolipoprotein A-I Cell adhesion: Vitronectin Others: A disintegrin and metalloproteinase with thrombospondin motifs 12	[20]
Riboflavin-coated SPIONs	Cores made of iron oxides (e.g., magnetite or maghemite)	Theranostic applications	Complement pathway: Complement C4 (fragments) Complement alternate pathway: Complement factor H Lipid transport: Apolipoprotein E, apolipoprotein A-I Others: Hemoglobin fetal subunit beta, hemoglubin subunit alpha, serum albumin, peptidyl-prolyl cis-trans isomerase A, tetranectin, α-2-HS-glycoprotein	[21]
Carboxylated polystyrene-NPs	Polystyrene, surface carboxyl groups	Drug delivery and diagnostic fields	Blood coagulation: Fibrinogen, histidine-rich glycoprotein, kininogen-1, plasma kallikrein Adaptive immunity: Immunoglobulin Complement pathway: Complement components, clusterin Lipid transport: Apolipoproteins Cell adhesion: Vitronectin Others: Serum albumin, trypsin inhibitor heavy chains, beta-2-glycoprotein 1	[22]
TiO_2_ NPs	Titanium dioxide	Nanoparticle toxicity studies	Host–virus interaction: Moesin, annexin A2, keratin (type II cytoskeletal 8) Autophagy: Ras-related protein Rab-8A Others: Pulmonary surfactant-associated protein A1, actin (cytoplasmic 1), L-lactate dehydrogenase A-like 6A, alpha-actinin-4, POTE ankyrin domain family member E, serum albumin	[23]

^a^ Proteins were grouped into representative biological processes (blood coagulation, complement pathway, adaptive immunity, lipid transport, and others) using gene ontology. ^b^ For the formation of protein coronas, nanoparticles were incubated in: FVB/N mouse plasma (liposomes), female CD1 mice plasma (lysolipid-containing TSL), human plasma (PEGylated cationic liposomes, silica NPs modified with surface NH_2_, silica NPs bioconjugated with PEG and transferrin, carbon nanotubes and carboxylated polystyrene-NPs), fetal bovine serum (negatively charged hydrophilic silica NPs and riboflavin-coated SPIONs), and human bronchoalveolar lavage fluid from patients with protein alveolar proteinosis (TiO_2_ NPs). ^c^ Abbreviations: NPs, nanoparticles; PC, protein corona; HSPC, hydrogenated soy phosphatidylcholine; DSPG, 1,2-distearoylsn-glycero-3-[phospho-rac-(1-glycerol)] (sodium salt); Chol, cholesterol; TSL, temperature-sensitive liposomes; DPPC, dipalmitoylphosphatidylcholine; MSPC, monostearoyl phosphatidylcholine; HSPC, hydrogenated soy phosphatidylcholine; DSPE-PEG2000, 1,2-distearoyl-sn-glycero-3-phosphoethanolamine-N-[methoxy(polyethylene glycol)-2000; Dox, doxorubicin hydrochloride; DOTAP, 1,2-Dioleoyl-3-trimethylammonium-propane; DC-Chol, (3b-[N-(N0,N0-dimethylaminoethane)-carbamoyl])-cholesterol; DOPC, dioleoylphosphocholine; DOPE, dioleoylphosphatidylethanolamine; PEG, polyethyleneglycol; SWCNTs, single-walled carbon nanotubes; SPIONs, superparamagnetic iron oxide nanoparticles; PS, polystyrene; TiO_2_, titanium dioxide.

## Data Availability

Data on Appendix A is available upon request.

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
