# Peer review of "Could a Non-Cellular Molecular Interactome in the Blood Circulation Influence Pathogens’ Infectivity?"

_cells, 2023, doi:10.3390/cells12131699_

Round 1

Reviewer 1 Report

I would say that this article provides a comprehensive and detailed exploration of the interactions between pathogens and the cellular and non-cellular components of the blood. The article proposes that the non-cellular blood interactome may play a crucial role in innate immunity and in modulating pathogens' infectivity potential, and may be specific to the host.

The article discusses the formation of protein coronas on artificial nanoparticles and the possibility of viruses and bacteria acquiring biocoronas, which can influence their behavior and interactions with host cells. The article provides specific examples, such as SARS-CoV-2 and its Spike protein, and proposes new candidate therapeutic targets to improve treatment of viral and bacterial conditions. The article also presents a novel bait-prey thermal proteome profiling technique for identifying pathogen-specific molecular interactors in the blood.

Overall, this article presents valuable insights into the complex interactions between pathogens and the blood, and proposes new avenues for research and treatment.

Author Response

We much appreciate the constructive and insightful comments by all four reviewers. We have addressed each and every one of their questions and suggestions in a new section "Outlook". This new section of the manuscript, which is highlighted yellow, is located at the end of the revised manuscript. We think that these revisions have improved the manuscript.

Comment: I would say that this article provides a comprehensive and detailed exploration of the interactions between pathogens and the cellular and non-cellular components of the blood. The article proposes that the non-cellular blood interactome may play a crucial role in innate immunity and in modulating pathogens' infectivity potential and may be specific to the host.

The article discusses the formation of protein coronas on artificial nanoparticles and the possibility of viruses and bacteria acquiring biocoronas, which can influence their behavior and interactions with host cells. The article provides specific examples, such as SARS-CoV-2 and its Spike protein, and proposes new candidate therapeutic targets to improve treatment of viral and bacterial conditions. The article also presents a novel bait-prey thermal proteome profiling technique for identifying pathogen-specific molecular interactors in the blood.

Overall, this article presents valuable insights into the complex interactions between pathogens and the blood, and proposes new avenues for research and treatment.

Response: Thank you for kind words on our article.

Reviewer 2 Report

The manuscript aims to describe the formation of non-cellular interactome or biocorona on the surface of nano/micro sized entities making a parallel between nanoparticles, viruses and bacteria. Moreover, the authors describe how this interactome can influence the infectivity of pathogens. The idea is not so original but very important in order to understand the mechanisms of interaction between the host and pathogen and between cells and nanoparticles used in nanomedicine. The review summarises and describes very well the topic and the authors embellish the concepts by discussing their interesting evidences on the interaction between SARS-Cov 2 and cortisol/glucocorticoids. Thus, I do not observe relevant limitations.

However, I would like to suggest the authors to stress a little bit more the importance of the non-cell interactome to be responsible for the differences in the susceptibility to infections as it is highly individual-specific. Moreover, I suggest to add some commments and discussion on the possibility to have different non-cell interactomes based on the routes of the administration of nanoparticles (e.g., oral ingestion, inhalation, dermal penetration, intravascular injection) as well as on the routes of the entry for pathogens (e.g., blood or mucosal tissue etc.). 

Author Response

We much appreciate the constructive and insightful comments by all four reviewers. We have addressed each and every one of their questions and suggestions in a new section "Outlook". This new section of the manuscript, which is highlighted yellow, is located at the end of the revised manuscript. We think that these revisions have improved the manuscript.

Comment 1: I would like to suggest the authors to stress a little bit more the importance of the non-cell interactome to be responsible for the differences in the susceptibility to infections as it is highly individual-specific.

Response: We now comment on this important suggestion in page 12, lines 473 to 498.

Comment 2: Moreover, I suggest adding some comments and discussion on the possibility to have different non-cell interactomes based on the routes of the administration of nanoparticles (e.g., oral ingestion, inhalation, dermal penetration, intravascular injection) as well as on the routes of the entry for pathogens (e.g., blood or mucosal tissue etc.).

Response: We have addressed this interesting suggestion in pages 13, line 540 on. In addition to addressing the reviewers' suggestion we now comment on the structure of the pathogen-attracted molecular interactome.

We sincerely thank the editor and the four reviewers for very helpful comments which have undoubtedly improved the quality of this manuscript.

Reviewer 3 Report

This manuscript is novel in that the entry of artificial nanomaterials into the bloodstream forms a "protein corona" with non-cellular components, analogous to the non-cellular aggregation of pathogens into the organism. A proteomic approach to bait-prey interactions combined with pathogen-specific infectivity tests is proposed for the rapid identification of pathogen-specific molecular interactors in blood.

1. Page 2, line 56. we develop the hypothesis that pathogens in the blood circulation encounter a non-cellular molecular interactome that may include large and small biomolecules (proteins, peptides, metabolites) which can be recruited to the pathogens’ surface (much like a biomolecular corona).

Question 1: When a pathogen enters the bloodstream, is it the immune cell recognition or the non-cellular molecular recruitment mentioned in the article that occurs first?

2. Page 2, line 60. We argue that the non-cellular blood interactome is likely specific to the host and capable of modifying the infectivity potential of pathogens; its characterization could help explain the differencesin susceptibility to pathogens’ infection among individuals as well as improve the effective clinical management of infectious and non-infectious conditions.

Question 1: Combined with extant studies, this suggests that pathogen susceptibility is enhanced in the process

3. Page 3, line 97. The formation of a protein corona on nanoparticles is mediated by mutual electrostatic and hydrophobic interactions, hydrogen bonding, van der Waals forces or π-π stacking stacking.

Question 1: How do pathogens containing biologically active substances differ from artificial nanoparticles in the way they form protein coronas?

4. Page 6, line 242. Subheading 2.3 needs to be modified according to the structure of the article.

5. Some literature related to bacterial infections should be cited (e.g., Bioactive materials, 2021, 6(12): 4389-4401; Nano Today, 2022, 43: 101380; Journal of materials chemistry B, 2020, 8(15): 3010-3015).

This manuscript is novel in that the entry of artificial nanomaterials into the bloodstream forms a "protein corona" with non-cellular components, analogous to the non-cellular aggregation of pathogens into the organism. A proteomic approach to bait-prey interactions combined with pathogen-specific infectivity tests is proposed for the rapid identification of pathogen-specific molecular interactors in blood.

1. Page 2, line 56. we develop the hypothesis that pathogens in the blood circulation encounter a non-cellular molecular interactome that may include large and small biomolecules (proteins, peptides, metabolites) which can be recruited to the pathogens’ surface (much like a biomolecular corona).

Question 1: When a pathogen enters the bloodstream, is it the immune cell recognition or the non-cellular molecular recruitment mentioned in the article that occurs first?

2. Page 2, line 60. We argue that the non-cellular blood interactome is likely specific to the host and capable of modifying the infectivity potential of pathogens; its characterization could help explain the differencesin susceptibility to pathogens’ infection among individuals as well as improve the effective clinical management of infectious and non-infectious conditions.

Question 1: Combined with extant studies, this suggests that pathogen susceptibility is enhanced in the process

3. Page 3, line 97. The formation of a protein corona on nanoparticles is mediated by mutual electrostatic and hydrophobic interactions, hydrogen bonding, van der Waals forces or π-π stacking stacking.

Question 1: How do pathogens containing biologically active substances differ from artificial nanoparticles in the way they form protein coronas?

4. Page 6, line 242. Subheading 2.3 needs to be modified according to the structure of the article.

5. Some literature related to bacterial infections should be cited (e.g., Bioactive materials, 2021, 6(12): 4389-4401; Nano Today, 2022, 43: 101380; Journal of materials chemistry B, 2020, 8(15): 3010-3015).

Author Response

We much appreciate the constructive and insightful comments by all four reviewers. We have addressed each and every one of their questions and suggestions in a new section "Outlook". This new section of the manuscript, which is highlighted yellow, is located at the end of the revised manuscript. We think that these revisions have improved the manuscript.

Comment 1: Page 2, line 56. we develop the hypothesis that pathogens in the blood circulation encounter a non-cellular molecular interactome that may include large and small biomolecules (proteins, peptides, metabolites) which can be recruited to the pathogens’ surface (much like a biomolecular corona).

Question 1: When a pathogen enters the bloodstream, is it the immune cell recognition or the non-cellular molecular recruitment mentioned in the article that occurs first?

Response: We have addressed this interesting question in the last paragraph of page 11, line 448 and in the following lines of the same paragraph in page 12 of the revised manuscript.

Comment 2: Page 2, line 60. We argue that the non-cellular blood interactome is likely specific to the host and capable of modifying the infectivity potential of pathogens; its characterization could help explain the differences in susceptibility to pathogens’ infection among individuals as well as improve the effective clinical management of infectious and non-infectious conditions.

Question 2: Combined with extant studies, this suggests that pathogen susceptibility is enhanced in the process.

Response: We have addressed this exciting question in page 12, line 473. 

Comment 3: Page 3, line 97. The formation of a protein corona on nanoparticles is mediated by mutual electrostatic and hydrophobic interactions, hydrogen bonding, van der Waals forces or π-π stacking stacking.

Question 3: How do pathogens containing biologically active substances differ from artificial nanoparticles in the way they form protein coronas?

Response: We have addressed this question in page 12, line 499.

Comment 4: Page 6, line 242. Subheading 2.3 needs to be modified according to the structure of the article.

Response 4: As requested, we have made changes to subheading 2.3.

Comment 5: Some literature related to bacterial infections should be cited (e.g., Bioactive materials, 2021, 6(12): 4389-4401; Nano Today, 2022, 43: 101380; Journal of materials chemistry B, 2020, 8(15): 3010-3015).

Response: As suggested, we have added these helpful references (numbered 11, 12, and 13) in the "References" section.

Reviewer 4 Report

This is a very interesting manuscript that explores the omics approach in pathogen invasive mechanisms, by assuming that the pathogen does not enter and circulate the host isolated, but in the complex interactome. The idea is deceptively simple and severely overlooked in past papers. It could actually serve to provide a step forward by explaining the gap between the host and pathogen side of the interaction. The manuscript is somewhat provocative, as it manages to demonstrate that several mechanisms and specific proteins participate in interactome creation. I would suggest two general pathways. The first one is complex, and that is the study design for the validation of these ideas. This is a major problem since it is difficult to test many of these hypotheses in conventional study. Maybe ex-vivo study, with selective removal of some components might be the best way forward, but a provocation phenotype will surely be needed in the experimental study design – all other approaches have an inherent bias that prevents any decent answers. The second part is that I would suggest the authors to focus more on glycans, which are known to interact and react in inflammation and infection. Are there similar studies published? If not, please raise this as a possible way forward, especially knowing that specific patterns of IgG may have an important role in disease susceptibility. I would also suggest that the authors consider spreading the idea of interactome to the disease severity, which also shows interesting and currently inexplicable patterns. Overall, a great starting point for further discussions and surely a manuscript worth publishing. 

Fine

Author Response

We much appreciate the constructive and insightful comments by all four reviewers. We have addressed each and every one of their questions and suggestions in a new section "Outlook". This new section of the manuscript, which is highlighted yellow, is located at the end of the revised manuscript. We think that these revisions have improved the manuscript.

Comment 1: I would suggest two general pathways. The first one is complex, and that is the study design for the validation of these ideas. This is a major problem since it is difficult to test many of these hypotheses in conventional study. Maybe ex-vivo study, with selective removal of some components might be the best way forward, but a provocation phenotype will surely be needed in the experimental study design – all other approaches have an inherent bias that prevents any decent answers.

Response: Thank you for this important observation which we have addressed in page 13, lines 514 to 525. Even though we know that a precise answer to this problem is beyond the scope of this paper, we have added some comments mainly on the issues related to traditional in vitro experiments.

Comment 2: The second part is that I would suggest the authors to focus more on glycans, which are known to interact and react in inflammation and infection. Are there similar studies published? If not, please raise this as a possible way forward, especially knowing that specific patterns of IgG may have an important role in disease susceptibility.

Response: We have addressed and commented on this key question in page 13, lines 526 to 539.

Comment 3: I would also suggest that the authors consider spreading the idea of interactome to the disease severity, which also shows interesting and currently inexplicable patterns.

Response: This is a significant question which we now address and comment on in page 13, lines 473-498.

Round 2

Reviewer 3 Report

The author made very detailed modifications according to my opinion. Therefore, I recommend that this manuscript be published in its current form.